# ReFrame: Layer Caching for Accelerated Inference in Real-Time Rendering

**Lufei Liu** [1]   **Tor M. Aamodt** [1]

## Abstract

Graphics rendering applications increasingly leverage neural networks in tasks such as denoising, supersampling, and frame extrapolation to improve image quality while maintaining frame rates. The temporal coherence inherent in these tasks presents an opportunity to reuse intermediate results from previous frames and avoid redundant computations. Recent work has shown that caching intermediate features to be reused in subsequent inferences is an effective method to reduce latency in diffusion models. We extend this idea to real-time rendering and present ReFrame, which explores different caching policies to optimize trade-offs between quality and performance in rendering workloads. ReFrame can be applied to a variety of encoder-decoder style networks commonly found in rendering pipelines. Experimental results show that we achieve 1.4× speedup on average with negligible quality loss in three real-time rendering tasks. Code available: https://ubc-aamodt-group.github.io/reframe-layer-caching/

## 1. Introduction

Real-time rendering is an important application that enables high-quality visualizations and interactivity across many industries such as gaming, virtual reality, design, and healthcare. In recent years, neural networks have become a critical component in real-time rendering workloads. To achieve high-quality renderings within a limited frame budget, only a small fraction of the image is rendered through traditional techniques like ray tracing. The majority of the pixels are actually generated using upsampling methods such as frame extrapolation and supersampling neural networks. The newest version of NVIDIA's Deep Learning Super Sampling (DLSS) technology, DLSS 4.0 (Lin & Burnes, 2025), applies neural networks to upscale images to 4× resolution and 4× frame rate, effectively rendering only one of every 16 pixels through traditional methods.

Real-time rendering applications are latency-sensitive, establishing an important trade-off between quality and performance. Neural network inferences now make up a significant portion of the rendering pipeline, especially with ray tracing accelerated using dedicated hardware like the NVIDIA RT Core. Any reduction of the inference latency can be used to allocate more resources to traditional rendering and improve the final image quality or directly improve the frame rate, both of which benefit user experience.

One approach to accelerate inference latency is by identifying sources of redundancy in the computation (LeCun et al., 1989). Rendering workloads naturally exhibit temporal coherence, where the content of consecutive frames is highly correlated (Scherzer et al., 2012). The inter-frame similarities are particularly high with fast frame rates, at which frame content changes slowly even with fast camera movements. These similarities also persist deep within the neural networks, exhibiting redundancy in the computation of intermediate layers. Figure 1 shows a sequence of 20 frames in an Unreal Engine demo scene to visually illustrate similarities between frames and slow scene changes.

A similar observation was recently made in the context of diffusion models, where intermediate layer outputs were cached and reused in subsequent inferences of the model by injecting cached results in concatenations with skip connections instead of computing deeper layers (Ma et al., 2024; Wimbauer et al., 2024). This caching scheme was shown to be effective in reducing the number of FLOPs required for inference, leading to faster inferences and lower computational costs. We explore adapting this caching technique to neural networks used in real-time rendering applications.

Although the caching technique was originally designed for U-Nets in diffusion models (Ho et al., 2020; Rombach et al., 2022), we show that it can be applied to a variety of encoder-decoder style networks commonly found in rendering pipelines. Adapting the technique for rendering workloads comes with several challenges. Unlike diffusion models that rely on long sequences of inference iterations to produce a single output, every inference iteration must produce a high-quality frame in real-time rendering. Therefore,

---

[1]Department of ECE, University of British Columbia, Vancouver, Canada. Correspondence to: Lufei Liu <liu-lufei@student.ubc.ca>, Tor M. Aamodt <aamodt@ece.ubc.ca>.

*Proceedings of the 42$^{nd}$ International Conference on Machine Learning*, Vancouver, Canada. PMLR 267, 2025. Copyright 2025 by the author(s).

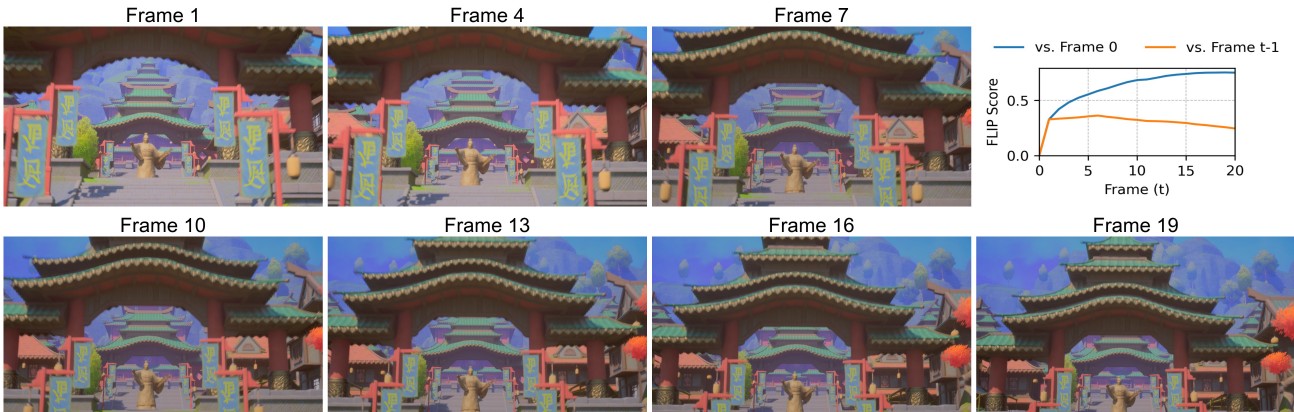

*Figure 1.* Input similarity in a rendering workload showing every three frames. Upper right plot shows the similarity compared to Frame 0 (Frame $t$ vs. Frame 0) and similarity between consecutive frames (Frame $t$ vs. Frame $t-1$) measured as FLIP scores.

approximate features in the cache used by ReFrame must trade image quality for improved frame rates in a way that does not impede the user experience. Furthermore, diffusion models rely on a fixed number of forward passes with a consistent pattern of change to guide the cache (Ma et al., 2024), while ReFrame must address the challenge of high variations and unpredictability in rendering workloads.

Our contributions are as follows:

- We extend existing methods of caching intermediate network features beyond U-Nets in diffusion models to a variety of encoder-decoder style networks that include feature concatenation.

- We explore new caching configurations to optimize trade-offs between quality and inference time that take advantage of patterns in real-time rendering workloads.

- We demonstrate empirically on an NVIDIA RTX 2080 Ti GPU that using ReFrame reduces the inference time of three different real-time rendering networks (frame extrapolation, supersampling, and image composition) by $1.4\times$ on average with negligible quality loss.

## 2. Related Work

This section reviews related research on exploiting frame similarity in rendering and video-processing, trading output accuracy for faster inference in neural networks, and the use of neural networks in real-time rendering applications.

### 2.1. Inter-frame Similarity

Temporal coherence in rendering workloads and in video processing has long been explored to reduce the computational cost of processing frames (Walter et al., 1999; Scherzer et al., 2012; Arnau et al., 2013). DeltaCNN (Parger

et al., 2022) and Event-NN (Dutson et al., 2022) exploit this similarity by processing only the differences (deltas) between frames of video from a still camera, effectively sparsifying the computation. MotionDeltaCNN (Parger et al., 2023) extends this idea to accommodate camera motion, and Eventful Transformers (Dutson et al., 2023) adapts it to transformer architectures. However, despite high FLOPs reduction, these techniques struggle to generate significant speedups because the underlying hardware is not optimized for the scattered nature of the eliminated computations, which is a problem that ReFrame avoids.

Hardware accelerators like Cambricon-D (Kong et al., 2024) and Diffy (Mahmoud et al., 2018) have also been proposed to exploit frame similarity on dedicated hardware, but are not adopted in commodity hardware. A more practical approach to exploiting frame similarity is through caching intermediate layer outputs of neural networks (Ma et al., 2024; Wimbauer et al., 2024; Xu et al., 2018), but has only been applied to diffusion models over long sequences of inferences. Caching approaches have also been used for neural rendering (Steiner et al., 2024), which is typically used for view synthesis rather than rendering synthetic virtual worlds. Inter-frame similarity is a well-known source of redundancy and ReFrame introduces a technique that produces real benefits in real-time rendering workloads.

### 2.2. Approximate Inference

Not all applications require exact inference results, and approximate inference can be used to reduce the computational cost of neural networks (Sun et al., 2024). Approximate inference can be achieved through quantization (Hubara et al., 2016), pruning (Han et al., 2016), or low-rank factorization (Denton et al., 2014) and demonstrate potential of trading accuracy for performance. These approaches recover the accuracy loss by training the network to adapt to

the approximate inference, while ReFrame is training-free. However, ReFrame is orthogonal to other approximate inference techniques and can potentially benefit from a combined approach with these methods.

## 2.3. Real-time Rendering Networks

In the real-time rendering pipeline, a low resolution image is usually generated through ray tracing, an algorithm capable of creating highly photo-realistic images. However, ray tracing is an expensive and very noisy process, and cannot independently produce a clear image within the frame budget to meet 60-90 frames per second. Therefore, a noisy low resolution ray-traced image is passed through a series of neural networks for denoising (Choi et al., 2024; Scardigli et al., 2024; Chen et al., 2023), frame extrapolation (Guo et al., 2021; Wu et al., 2023; Yang et al., 2024), and neural supersampling or super resolution (Zhang et al., 2024; He et al., 2024; Zhong et al., 2023; Yang et al., 2023; Mercier et al., 2023) to generate the final high resolution image. Augmented and virtual reality (AR/VR) applications further require additional network inferences for image composition (Watson et al., 2023; Yu et al., 2023). DLSS (Lin & Burnes, 2025) and Intel XeSS (Chowdhury et al., 2022) are popular examples of a neural networks that are widely used in the gaming industry for real-time rendering, but require high-end GPUs and can still benefit from dedicating more resources to a better input image.

Real-time rendering networks commonly take advantage of U-Net (Ronneberger et al., 2015) and U-Net++ (Zhou et al., 2018) architectures for their ability to capture spatial information at different scales, which is useful for extracting features from images and managing different image resolutions. Although these networks are designed to be fast, they still contribute a significant portion of the rendering pipeline latency, often exceeding capabilities of lower-end devices (Bhuyan et al., 2024; Yang et al., 2023). ReFrame aims to reduce FLOPs in these workloads to support better quality rendering on mobile devices.

## 3. ReFrame

We focus ReFrame on networks that include an encoder-decoder architecture with skip connections, such as U-Net or U-Net++, due to their ubiquity in state-of-the-art real-time rendering pipelines. These networks are characterized by a series of convolutional layers that downsample the input image ($I$) to a low-dimensional feature space, followed by a series of convolutional layers that upsample the feature space back to the original image resolution. At each downsampling block, the network branches off to a skip connection that concatenates to the corresponding upsampling block, illustrated in Figure 2 (a). For a U-Net with $n$ blocks, the output of the network ($O$) can be described as convolu-

tional blocks ($X^i$) that process outputs from previous blocks concatenated ($concat$) with skip connections:

$$O = X^n(concat(X^{n-1}(\\ concat(X^{n-2}(\cdots), X^1(X^0(I)))),\\ X^0(I))) \quad (1)$$

In a U-Net++, the network branches off to multiple skip connections at each downsampling block, creating a nested U-Net structure, as shown in Figure 2 (b). U-Net++ blocks are denoted as $X^{i,j}$, where $i$ is the depth of the block and $j$ indexes a convolution along the skip connection where $j \in [0, i] \cap \mathbb{Z}$.

## 3.1. Layer Caching

In a U-Net, the final output from the network is the result of the last block ($X^n$), which takes in a concatenation of results from previous blocks as its input. At the first frame (at time $t$), we compute the network inference end-to-end as normal, and store the inputs to the last block of the network in a cache ($C_t$). Specifically, we cache the portion that is produced by the previous encoder block, $X^{n-1}$, which contains high-level features extracted from the input image that often change slowly between frames. Then, for subsequent frames, we reuse the cached results ($C_t$) from the main branch and concatenate them with the skip connection from the first block, $X^0$. This process allows us to compute only $X^0$ and $X^n$, skipping all of the deeper layers in the network and replacing them with cached inferences, as described in DeepCache (Ma et al., 2024):

$$C_t = X^{n-1}(concat(X^{n-2}(\cdots), X^1(X^0(I)))) \quad (2)$$
$$O = X^n(concat(C_t, X^0(I))) \quad (3)$$

In a U-Net++ architecture, the final block ($X^{0,n}$) concatenates more skip connections than only the first block like in a U-Net. Each skip connections that feeds into a concatenation can be a potential candidate for caching. One possible way to extend the caching scheme used in DeepCache to U-Net++ is to cache the results from each branch, except the skip connection from the first block ($X^{0,0}$):

$$O = X^{0,n}(concat(C_t^{0,n-1}, C_t^{0,n-2}, \cdots, X^{0,0}(I))) \quad (4)$$

However, to better pass new information through the network, we instead cache the branches that lead into all the higher-level blocks ($X^{0,j}$). Figure 2 shows the U-Net and U-Net++ architectures with layer caching for the full inference frame and subsequent cached frames.

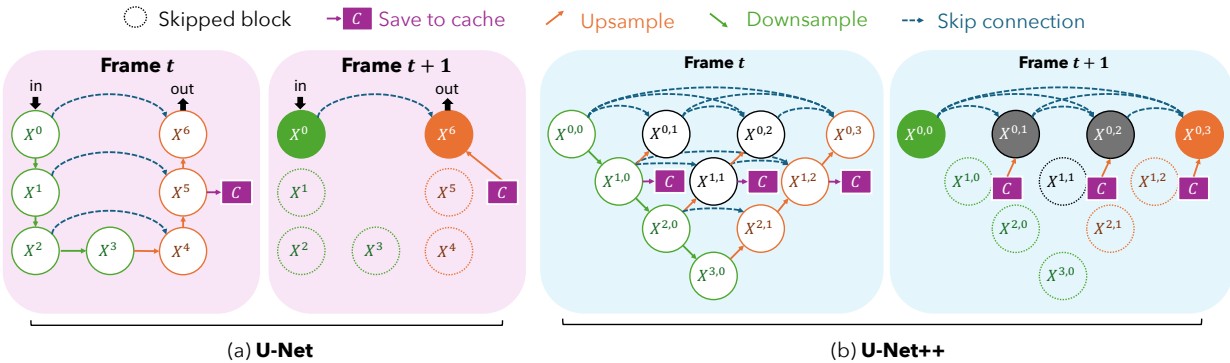

*Figure 2.* Diagrams of U-Net (a) and U-Net++ (b) architectures with layer caching. In frame $t$, the network computes the full inference end-to-end, saving intermediate outputs into a cache. In subsequent frames $t + 1$, the network reuses the cached results instead of re-computing the intermediate layers.

In some cases, ReFrame can be applied on networks beyond U-Net and U-Net++ architectures if the network comprises concatenations of intermediate layer outputs. For example, some networks concatenate results from several feature extraction blocks before passing them to the final block. In these cases, we can cache the results of any slow-changing feature extraction blocks to reuse in subsequent frames.

### 3.2. Cache Policies

Although the caching scheme is simple, the choice of which layers to cache and how to manage the cache can have a significant impact on the performance of the network. First, our caching scheme can be applied at any depth in the U-Net and U-Net++ architectures instead of just the last block. Figure 3 shows the features at different depths (levels) over consecutive frames in a rendering workload. Caching the last block is the most effective for reducing the number of FLOPs required, but also suffers the highest quality degradation because the changes in intermediate features (light green regions in Figure 3) at all levels are ignored. We find empirically that the quality degradation from caching at the last block is outweighed by the performance gains, which we demonstrate in Section 4.5. More importantly, we explore different policies to manage how the cache is refreshed. If the cache is updated every frame, the network will always compute the full inference, which defeats the purpose of the caching scheme. However, if the cache is never updated, the high-level features in the cache will become stale and no longer be representative of the input image. Ideally, the cache should be refreshed just before its contents noticeably affect the output quality.

#### 3.2.1. EVERY-N

DeepCache (Ma et al., 2024) proposes a simple policy to update the cache every $N$ frames, where $N$ can be a hyperparameter tuned to balance quality and performance. Although

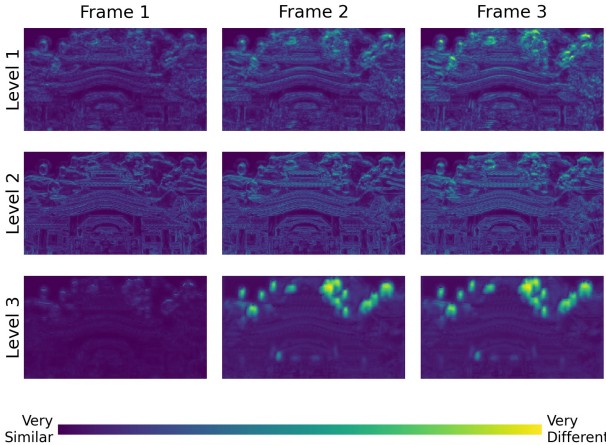

*Figure 3.* Feature similarity between consecutive frames in a rendering workload relative to Frame 0. Levels 1-3 correspond to the depth of the U-Net architecture (i.e. outputs of $X^0$, $X^1$, and $X^2$).

this policy is simple and effective, it fails to consider the variations in camera motion and scene complexity of real-time rendering workloads. For example, a fast-moving camera or a scene with many moving objects can quickly cause the cache to become stale. This behavior does not exist in diffusion models because the diffusion process is known to be smooth over time and not prone to sudden changes. DeepCache also proposed a non-linear cache policy that distributes the cache refreshes to fit temporal similarity patterns observed in diffusion models. The proposed distribution prioritizes refreshing the cache at the beginning and end of the image generation process where the most significant changes occur, but there is no such pattern for real-time rendering workloads. We evaluate both policies in Section 4.5 and compare it to adaptive policies for ReFrame.

### 3.2.2. FRAME DELTAS

We propose an adaptive policy that updates the cache only when the input image changes significantly from the cached frame. Instead of storing only $C_t$, we also store $I_t$, which we use to compute the symmetric mean absolute percentage error (SMAPE) between the input image and the cached image, $SMAPE(I, I_t)$. If the SMAPE exceeds a threshold $\tau$, we refresh the cache by computing the full inference and saving the new $C_{t'}$ and $I_{t'}$. This approach incurs additional computation for the SMAPE and additional memory to store $I_t$ but better adapts to the unpredictable nature of real-time rendering workloads, prevents sudden quality drops, and still reduces inference time overall as shown in Section 4.2. A secondary benefit is that the computed frame deltas can exploit techniques such as DeltaCNN (Parger et al., 2022) to further reduce the number of FLOPs required for inference.

By applying an adaptive policy, we increase average computation savings during periods of low camera motion or low scene complexity because the cache can be refreshed less frequently than in the every-N approach. Also, this approach avoids sudden quality drops than can occur with every-N policies when content changes are not aligned with the fixed refresh intervals. This benefit is particularly important in real-time rendering workloads because the end user may not notice smooth quality differences, but will notice sudden changes in quality (Thakolsri et al., 2011). We consider both high and low sensitivity thresholds (Delta_H and Delta_L) for the SMAPE in our experiments, which can be tuned to balance quality and performance.

### 3.2.3. MOTION VECTOR THRESHOLD

Another possible policy for real-time rendering is the use of motion vectors to identify when the input image changes significantly. Motion vectors are already rendered as part of the G-buffers in many rendering pipelines and are often required as an input for supersampling and frame extrapolation tasks. Therefore, we propose to set a maximum motion threshold $\tau$, then check whether the average motion exceeds $\tau$ at each frame, and refresh the cache if the threshold is exceeded. This policy results in similar advantages of adaptability as using frame deltas, but does not require any additional storage when motion vectors are already computed. However, motion is only one source of change in the input image, and frame deltas present a more comprehensive indicator for cache refreshes.

### 3.3. Quality Trade-offs

Our caching scheme introduces an additional trade-off between quality and performance, where the quality of the inference result is diminished due to the reuse of cached features. Caching at deeper layers in the network can help recover some of the quality degradation, but also reduces

the performance gains. A different approach for mitigating quality degradation is to allocate the saved inference time to rendering a better input image, such as increasing the samples per pixel count in ray tracing or slightly increasing the resolution of the input image. Depending on the application, this approach can lead to higher quality images overall or higher frame rates with negligible quality loss. We explore this trade-off between allocating resources to traditional rendering versus network inferences empirically in Section 4.2.2

## 4. Experiment Evaluation

We evaluate ReFrame on three common real-time rendering tasks: frame extrapolation (FE), supersampling (SS), and image composition (IC). In a rendering pipeline, all of these tasks may be required, in addition to tasks including ray tracing, denoising, adaptive sampling, and other processing steps. For each task, we execute end-to-end inference on a sequence of frames, comparing the quality and performance of ReFrame against the baseline network. We modify each network in PyTorch to add our caching scheme, following implementation details in Appendix A.2. We choose frame deltas with both high (Delta_H) and low (Delta_L) sensitivities as the default policy to refresh the cache for all experiments except the ablation study in Section 4.5, where we compare different cache policies.

### 4.1. Evaluation Metrics

We evaluate the performance of ReFrame using a combination of image quality metrics and latency metrics. For image quality, we use FLIP (Andersson et al., 2020), learned perceptual image patch similarity (LPIPS) (Zhang et al., 2018), structural similarity (SSIM), and peak signal to noise ratio (PSNR), which are all full-reference image quality assessments (FR-IQA) commonly used to evaluate rendering quality. The FLIP score is specifically designed to evaluate rendered images and measures the perceptual similarity between two images, which provides the most insight to how our technique affects the end-user experience. We use the results of the baseline network without ReFrame as the reference image for all metrics unless otherwise specified.

For latency, we compare relative speedup in inference latency measured on a NVIDIA RTX 2080 Ti GPU. We also measure the number of floating point operations (FLOPs) required for inference within each encoder-decoder network (Enc-Dec FLOPs) to quantify the computational savings of our caching scheme. We also report the proportion of frames that use the cached features to reduce computation (Skipped Frames). The remaining frames require full inference due to cache refreshes or sudden movements that invalidate the cache, and these frames might benefit from orthogonal techniques like DeltaCNN (Parger et al., 2022).

| Policy | Workload | Scene | Skipped Frames ↑ | Eliminated Enc-Dec FLOPs ↑ | Speedup ↑ | FLIP ↓ | SSIM ↑ | PSNR ↑ | LPIPS ↓ | MSE ↓ |
|--------|----------|-------|------------------|---------------------------|-----------|--------|--------|--------|---------|-------|
| *Delta_H* | FE | Sun Temple | 50% | 27% | 1.42 | 0.0169 | 0.994 | 41.41 | 0.0066 | 1.63 |
| | | Cyberpunk | 30% | 16% | 1.10 | 0.0207 | 0.981 | 31.08 | 0.0103 | 3.66 |
| | | Asian Village | 35% | 19% | 1.24 | 0.0241 | 0.985 | 33.19 | 0.0248 | 3.89 |
| | SS | Sun Temple | 40% | 29% | 1.30 | 0.0490 | 0.970 | 53.13 | 0.0263 | 17.62 |
| | IC | Garden Chair | 13% | 6% | 1.05 | 0.0006 | 1.000 | 47.93 | 0.0001 | 0.07 |
| *Delta_L* | FE | Sun Temple | 80% | 43% | 1.72 | 0.0325 | 0.984 | 37.35 | 0.0198 | 4.71 |
| | | Cyberpunk | 60% | 32% | 1.49 | 0.0345 | 0.968 | 31.29 | 0.0181 | 6.07 |
| | | Asian Village | 60% | 32% | 1.55 | 0.0462 | 0.969 | 32.44 | 0.0508 | 7.81 |
| | SS | Sun Temple | 80% | 57% | 1.85 | 0.1180 | 0.930 | 34.52 | 0.0725 | 41.40 |
| | IC | Garden Chair | 79% | 34% | 1.20 | 0.0127 | 0.991 | 33.34 | 0.0109 | 1.79 |

*Table 1.* Performance and image quality results of our selected workloads (FE - frame extrapolation, SS - supersampling, IC - image composition). We compare two sensitivity settings in the delta cache policy: high sensitivity (*Delta_H*) and low sensitivity (*Delta_L*), detailed in Appendix A.3. Results are relative to the baseline network without caching. Enc-Dec FLOPs ↑ do not include additional inference operations outside the encoder-decoder network.

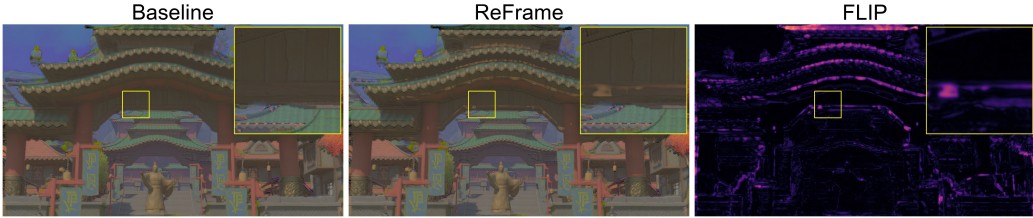

*Figure 4.* Frame extrapolation results with and without caching on the Asian Village scene using ExtraNet. The FLIP error map is shown on the right (pink indicates highly noticeable regions of differences).

## 4.2. Performance and Quality Results

Table 1 summarizes our key results, showing that ReFrame consistently reduces FLOPs and inference latency across all tasks. Our FLIP score results highlight that perceptual quality loss is negligible. All scores are generally below the acceptable losses observed in other neural rendering systems, which report scores between 0.05 and 0.28 in their final results (Müller et al., 2021; Li et al., 2022; Vaidyanathan et al., 2023). Even with higher MSE in some workloads, user experience should not be affected, as demonstrated by low perceptual loss scores such as FLIP and LPIPS.

ReFrame applies to 72% of inferences on average with a low sensitivity setting (Delta_L), resulting in a 40% reduction in FLOPs and 1.6× speedup in inference latency. With a high sensitivity setting (Delta_H), our caching scheme still reduces 19% of FLOPs, with negligible quality loss. Even though the cache is only applied to a subset of inferences on a portion of the network, we still achieve performance gains in all cases.

### 4.2.1. FRAME EXTRAPOLATION

Frame extrapolation takes historical and current frames as inputs (ex. $I_{t-2}$, $I_{t-1}$, and $I_t$) and predicts future in-between frames (ex. $I_{t+0.5}$). The final result displayed to the user alternates between rendered frames and extrapolated frames, which doubles the effective frame rate. We evaluate our caching scheme on ExtraNet (Guo et al., 2021), which uses a U-Net architecture with skip connections as a major component of the network. We generate our test set with the Unreal Engine build published by the authors and free scenes from the online Unreal Engine Marketplace (Fab), detailed in Appendix A.1. Although the scenes we evaluate are not identical to the original dataset, which is not freely available, we are only interested in the relative performance between the baseline and cached networks so this does not affect our evaluation. Figure 4 shows a visual example of our frame extrapolation results with and without caching, including the FLIP error map.

### 4.2.2. NEURAL SUPERSAMPLING

Neural supersampling is similar to image super resolution and upsamples a low resolution (LR) rendering input to a high resolution (HR) image output, but supersampling is adapted to reflect aliasing properties in computer graphics (Xiao et al., 2020). We evaluate ReFrame on a Fourier-based super resolution network (FBSR) for supersampling (Zhang et al., 2024). The network concatenates

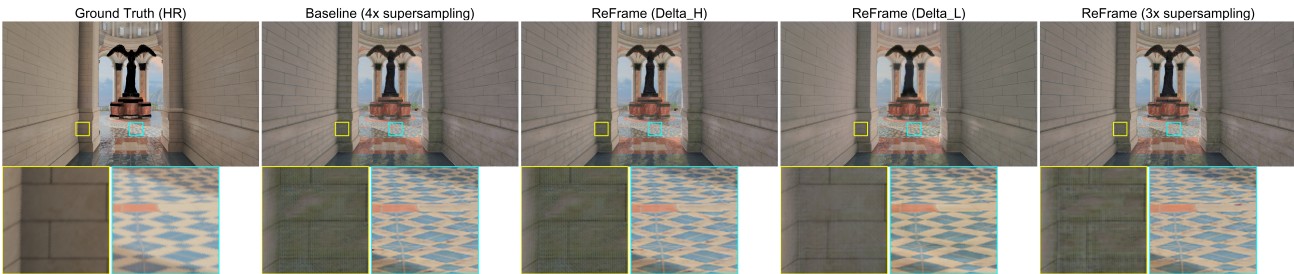

*Figure 5.* Supersampling results on the Sun Temple scene using FBSR. Ground truth image is rendered at high resolution. Delta_H and Delta_L show results with caching enabled. 3× supersampling shows quality improvements from using a larger input image. Zoomed-in regions show detailed quality differences between each configuration.

| Scaling Factor | Config | Total Time (s) ↓ | Time Savings ↑ | FLIP ↓ | SSIM ↑ | PSNR ↑ | LPIPS ↓ |
|---|---|---|---|---|---|---|---|
| 4× | Baseline | 1.459 | 0.00 | 0.398 | 0.761 | 21.305 | 0.356 |
| | Delta_H | 1.122 | 0.34 | 0.399 | 0.761 | 21.239 | 0.357 |
| | Delta_L | 0.788 | 0.67 | 0.401 | 0.761 | 21.040 | 0.367 |
| 3× | Baseline | 1.506 | -0.05 | 0.395 | 0.765 | 21.353 | 0.359 |
| | Delta_H | 1.27 | 0.19 | 0.396 | 0.766 | 21.316 | 0.360 |
| | Delta_L | 0.932 | 0.53 | 0.399 | 0.761 | 21.174 | 0.371 |

*Table 2.* Supersampling results on the Sun Temple scene. The scaling factor is the ratio of 1080p output resolution to 270p and 360p input resolutions. Time savings is measured relative to the baseline inference time for 4× scaling. 3× scaling is slower than 4× scaling without ReFrame. Image quality is measured against ground truth 1080p rendered images.

three feature extraction blocks that feed into a feature fusion block. Similar to caching downsampling blocks in the U-Net architecture, we cache the temporal and HR feature extraction blocks in FBSR. We use the same Unreal Engine build and the Sun Temple scene from our test set to evaluate caching for FBSR. Figure 5 shows a visual example of our supersampling results with different cache policies compared to the baseline and ground truth HR images.

We also evaluate the trade-off between quality and performance by comparing between a small LR input using no cache and a larger LR input with a cache. Table 2 shows that by using a cache, we can improve the final image quality by allocating the saved inference time to rendering a larger low resolution image. Although inference time for 3× upscaling is slower (-0.05s) due to the larger input size, by applying our proposed caching scheme, we can exploit the quality benefits of the larger input without suffering performance penalties. For example, 3× upscaling with a cache achieves a higher FLIP score with the Delta_H policy and results in 0.19s latency savings over 20 frames.

#### 4.2.3. IMAGE COMPOSITION

Image composition is useful for augmented reality (AR) applications that require multiple images to be combined into a single frame. We choose Implicit Depth (Watson

et al., 2023) as our test network, which estimates the depth of a real image and composites a virtual rendering into the real scene with correct occlusions. Implicit Depth uses a U-Net++ architecture for depth estimation and we evaluate the network using sample data released by the authors. Figure 6 shows a visual example of our image composition workload, with the FLIP error map comparing the final composite results with and without caching.

Figure 7 shows the inference time of the decoder blocks in Implicit Depth over time. By applying our caching scheme, the average inference time is significantly reduced. Spikes in inference time are cache refreshes, which occur when the input image changes significantly from the cached frame and only add a small latency overhead. Using our Delta_L policy, we can store the cache for longer periods of time when the input image is stable, as shown in the first 20 frames. Then, when the input image shows more changes, the cache is refreshed more often to prevent quality degradation.

### 4.3. Overheads

The performance overheads of storing and loading cached tensors are negligible compared to the performance gains from reducing computation. There is a small lag during cache refreshes to compute frame deltas as observed in Figure 7 for Delta_L, but this additional latency falls in the range of noise for the full inference time and can be eliminated with simpler policies such as every-N. Also, the memory overhead of our caching scheme is small, storing only one or a few tensors in total for the cache and one copy of the network input for frame deltas. Details of the cache memory consumption are included in Appendix A.6.

### 4.4. Comparison to DeltaCNN

DeltaCNN (Parger et al., 2022) is another method that exploits frame similarity by processing only the differences (deltas) between frames. We evaluate DeltaCNN on the frame extrapolation task using the Asian Village scene from our test set and the public DeltaCNN library for PyTorch. Ta-

Real      Render      Composite      Composite w/ ReFrame      FLIP

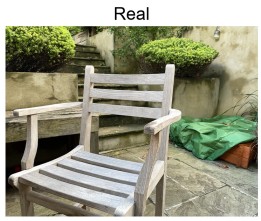 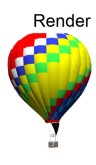 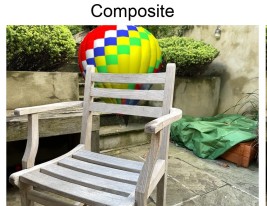 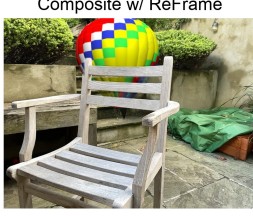 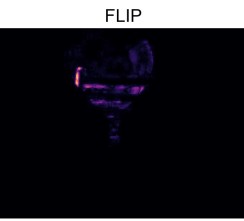

*Figure 6.* Implicit Depth results combining Real and Render into Composite. The FLIP error map is shown on the right comparing results with and without using a cache, where pink indicates highly noticeable regions of differences.

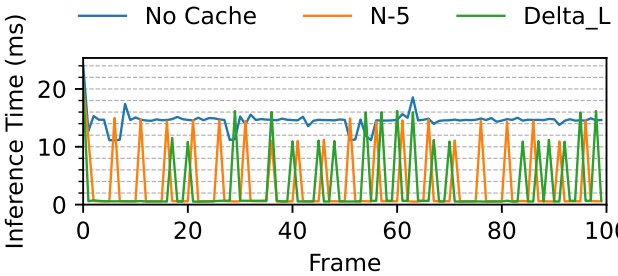

*Figure 7.* Inference time of decoder blocks in Implicit Depth over 100 frames. Both N-5 and Delta_L policies show significant reductions in inference time compared to No Cache. Delta_L spreads cache refreshes more strategically to enforce output quality.

| Frame | Theoretical FLOPs ↓ | Actual FLOPs ↓ | FLIP ↓ | SSIM ↑ | PSNR ↑ | LPIPS ↓ | MSE ↓ |
|---|---|---|---|---|---|---|---|
| 0 - Ref | 49.954 | 50.098 | 0 | 1.000 | - | 0 | 0 |
| 1 - Delta | 44.124 | 48.561 | 0.130 | 0.854 | 28.413 | 0.148 | 37.502 |
| 2 - Delta | 41.788 | 46.449 | 0.184 | 0.826 | 25.909 | 0.200 | 69.130 |

*Table 3.* Results of ExtraNet with DeltaCNN on Asian Village scene. Full inference is computed in the first frame (0 - Ref), then only deltas for the subsequent frames (1 - Delta and 2 - Delta). FLOPs are reported using the DeltaCNN performance metrics manager. Actual FLOPs are higher than theoretical savings due to the overhead of the DeltaCNN framework and tiling strategy.

ble 3 reports our results and show that our caching scheme is more effective in reducing FLOPs and improving inference latency than DeltaCNN. Techniques such as DeltaCNN require very high sparsity to achieve realized speedups unless a dedicated sparse accelerator is used, which is not commercially available. Furthermore, DeltaCNN is designed for video processing, which typically take a single RGB frame as input, while our rendering workloads require several concatenated G-buffers and time-warped frames as input. Unfortunately, these input configurations are less suitable for computing deltas between frames, worsened by the channel-wise masking strategy used in DeltaCNN.

Our caching scheme can be combined with DeltaCNN to further reduce FLOPs in the non-cached blocks of the network. We evaluate the combined method on the frame extrapolation task but find that there is little performance improve-

|  | Level 3 | Level 2 | **Level 1** | U-Net++ A | **U-Net++ B** |
|---|---|---|---|---|---|
| FLIP ↓ | 0.023 | 0.026 | 0.035 | 0.013 | 0.014 |
| SSIM ↑ | 0.982 | 0.979 | 0.968 | 0.991 | 0.989 |
| PSNR ↑ | 34.04 | 33.37 | 31.29 | 33.34 | 31.03 |
| LPIPS ↓ | 0.010 | 0.012 | 0.018 | 0.011 | 0.013 |
| MSE ↓ | 3.457 | 4.087 | 6.073 | 1.787 | 1.918 |
| FLOPs ↓ | 71.8% | 62.9% | 46.2% | 56.6% | 30.1% |

*Table 4.* Results of ablation study on ExtraNet and Implicit Depth comparing different cache levels and configurations. Image quality results are averaged over all frames in the test set and FLOPs ↓ are relative to the baseline encoder-decoder network.

ment versus using ReFrame only. Although the combined method reduces FLOPs, the hardware is not optimized for the scattered nature of the eliminated computations, causing high overheads and low performance gains.

### 4.5. Ablation Study

We choose to cache the last block of the U-Net architecture for all experiments because it is the most effective for reducing the number of FLOPs required. We explore caching deeper blocks in the U-Net architecture, which reduces fewer FLOPs but also suffers less quality degradation. Similarly, the U-Net++ architecture can be cached in a variety of ways, with each concatenation in the network offering a potential caching point. We choose to cache at the last block (Config B), but also consider caching all additional U-Net++ blocks while only evaluating the foundational U-Net blocks during cached inference (Config A). More details on the configurations are provided in Appendix A.3. Table 4 presents the results of our ablation study, showing that all options result in low quality degradation. Figure 8 visually shows that errors are nearly unnoticeable and indistinguishable between levels according to the FLIP error map. Therefore, we choose the option that produces the most significant performance gains.

We also evaluate other cache policies on the frame extrapolation task, including updating the cache every $N = 2$ frames (N-2), every $N = 5$ frames (N-5), based on motion vectors, and based on the non-linear cache policy proposed by DeepCache (Ma et al., 2024) (Non-Linear N-5). Table 5

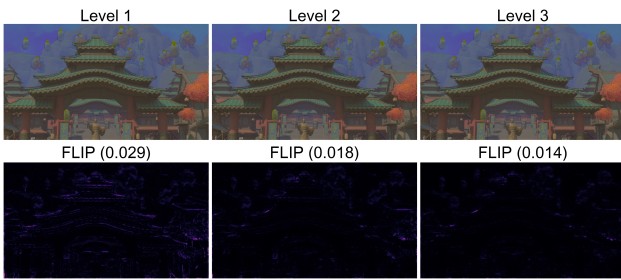

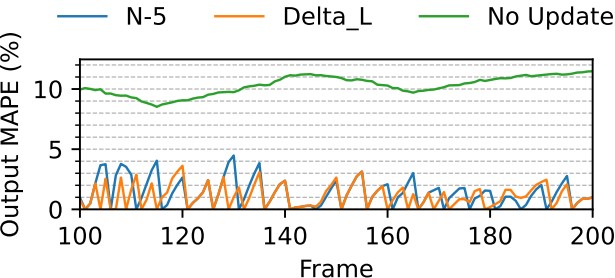

*Figure 8.* Ablation study results for ExtraNet U-Net architecture. Config A caches all U-Net++ blocks, while Config B caches only the last block.

| Policy | Delta_L | Delta_H | N-5 | N-2 | Motion Vector | Non-Linear N-5 |
|---|---|---|---|---|---|---|
| # Refresh Frames | 3 | 6 | 2 | 5 | 4 | 2 |
| FLIP ↓ | 0.035 | 0.021 | 0.040 | 0.025 | 0.030 | 0.040 |
| SSIM ↑ | 0.968 | 0.981 | 0.963 | 0.976 | 0.971 | 0.961 |
| PSNR ↑ | 31.29 | 31.08 | 31.39 | 31.34 | 31.22 | 31.21 |
| LPIPS ↓ | 0.018 | 0.010 | 0.020 | 0.013 | 0.016 | 0.022 |
| MSE ↓ | 6.073 | 3.655 | 7.046 | 4.653 | 5.471 | 7.259 |

*Table 5.* Results of ablation study on ExtraNet comparing different cache refresh policy settings on a sequence of 10 frames. Refresh frames are full inference computations to refresh the cache. Delta_L results in better image quality with fewer cache refreshes.

shows that the frame delta policy (Delta_L) outperforms the other policies in balancing performance and quality. Motion vectors are also very effective, but are not reliably available in all rendering workloads. Figure 9 confirms that the every-N policy suffers from spikes of low quality when movements do not match the cache refresh rate, while the maximum error from Delta_L is consistently lower. Figure 7 also shows that the Delta_L policy allocates cache refreshes more strategically according to the input behavior, leading to results with more consistent quality.

## 5. Discussion and Limitations

ReFrame is designed for U-Net and encoder-decoder style networks only and does not support other architectures such as transformers. Although there are many transformer-based models used in real-time rendering applications, such as DLSS 4.0 (Lin & Burnes, 2025), we believe U-Net-like convolutional networks are still heavily employed and more feasible to execute on lower-end devices. ReFrame is particularly useful on these lower-end devices where trading slight quality loss for latency reduction is valuable. The trade-off between quality and latency will differ for each network, depending on the cache configuration and the portion of the overall network architecture that can be skipped.

Although ReFrame can effectively reduce inference time in many frames to lower average latency, our technique cannot maintain a consistently faster frame rate since the cache

*Figure 9.* Mean absolute percentage error (MAPE) of outputs from Implicit Depth using N-5, Delta_L, and No Update cache policies after 100-frame warmup. Both N-5 and Delta_L policies show low errors, but Delta_L avoids larger error spikes.

will require periodic refreshes. Since ReFrame focuses on the post-processing stage that enhances the rendered images, additional image quality improvements from the neural networks can be included in a best-effort manner. Furthermore, reducing computation on average still reduces energy consumption, which is an especially important target in mobile-class devices such as VR headsets.

## 6. Conclusion

In this work, we propose a novel cache mechanism for neural networks that exploits the temporal coherence of real-time rendering workloads. Our method caches intermediate layer outputs in encoder-decoder style networks and reuses them in subsequent frames to reduce the number of FLOPs required for inference. Our proposed approach is limited to networks with skip connections and concatenations between intermediate layers, but this is a common architecture in real-time rendering pipelines. We explore different cache refresh policies and demonstrate reduced inference latency on three real-time rendering tasks. As future work, we plan to investigate more sophisticated cache policies with dynamic sensitivity settings and applying our method to other network architectures.

## Acknowledgements

We thank the reviewers for their feedback and the authors of our test workloads for their open-source implementations. This work was supported by the Natural Sciences and Engineering Research Council of Canada (NSERC). Tor Aamodt recently served as a consultant for IBM and Cisco.

## Impact Statement

This paper presents work whose goal is to advance the field of Machine Learning. There are many potential societal consequences of our work, none which we feel must be specifically highlighted here.

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

# A. Appendix

## A.1. Test Set

We select free scenes from the online Unreal Engine Marketplace to generate our test set and render with a modified version of Unreal Engine 5.1 from the authors of ExtraNet (Guo et al., 2021) to produce necessary G-buffer data. Table 6 summarizes the test scenes used for evaluation.

| Scene | Sun Temple | Cyberpunk | Asian Village |
|---|---|---|---|
| | | | |
| **Resolution** | 1920x1080 | 1920x1080 | 1920x1080 |
| **Test frames** | 20 | 10 | 20 |
| **Frame sequence** | 150 | 30 | 150 |
| **Description** | Scene from NVIDIA ORCA repository | Scene from video game Cyberpunk 2077 | Free scene from Unreal Engine Marketplace |
| **Buffers** | Base color, normal, metallic, roughness, scene depth, motion vector, HDR color, world position, NoV | | |

*Table 6.* Test scenes used for evaluation.

Authors of Implicit Depth (Watson et al., 2023) released their sample test scenes, which we use to evaluate the performance of ReFrame on their network. The Garden Chair scene has 295 frames and is rendered at $720 \times 540$ resolution.

## A.2. Cache Implementation

In this section, we provide more details on how we adapted our test networks to include the caching scheme.

### A.2.1. EXTRANET

ExtraNet (Guo et al., 2021) uses several history encoders to capture temporal information, which are concatenated to a U-Net architecture. We cache the U-Net as described in Section 3.1, which includes the inputs from the history encoders.

### A.2.2. IMPLICIT DEPTH

Implicit Depth (Watson et al., 2023) uses SimpleRecon (Sayed et al., 2022) as the backbone network to estimate depth features. SimpleRecon is based on a U-Net++ architecture, using ResNet blocks for the encoder and a nested U-Net for the decoder. We cache the U-Net++ architecture as described in Section 3.1 and keep the remaining network the same.

### A.2.3. FBSR

Fourier-based super resolution (FBSR) (Zhang et al., 2024) does not use a standard U-Net or U-Net++ architecture. However, the network still generally follows a decoder-encoder structure, with feature extraction blocks concatenated before the final block. We apply a similar principle and cache inputs to the concatenation to be reused in subsequent frames. In this case, we choose to cache the temporal and high resolution (HR) features since the low resolution (LR) features are most similar to the shallow layers in a U-Net and provides the most contextual information of the current frame. We also consider caching only the temporal features, but find that while quality does not improve noticeably, the performance gains are not as significant as caching both temporal and HR features. Figure 10 shows the FBSR architecture with ReFrame.

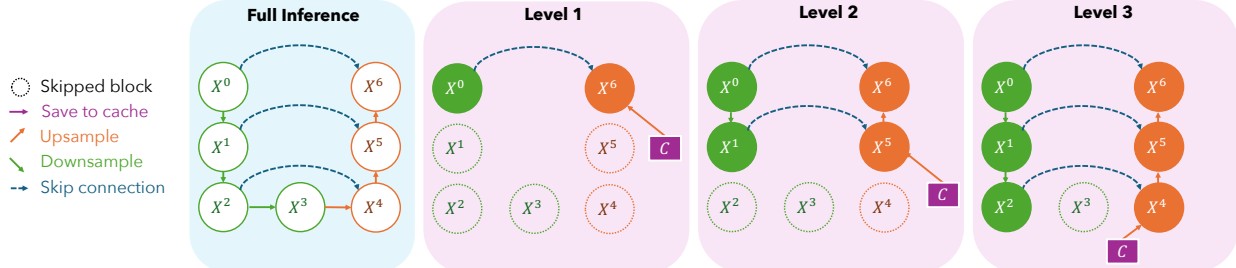

*Figure 10.* Diagrams of the FBSR architecture with ReFrame. In frame $t$, the network computes the full inference end-to-end, saving intermediate features from the temporal recurrent feature extractor and HR G-buffer feature extractor. In subsequent frames $t + 1$, the network reuses the cached features.

### A.3. Ablation Configurations

Figure 11 shows the Level 1-3 cache configurations used in the ablation study for U-Net architectures. Figure 12 shows the U-Net++ A and B cache configurations used in the ablation study for U-Net++ architectures.

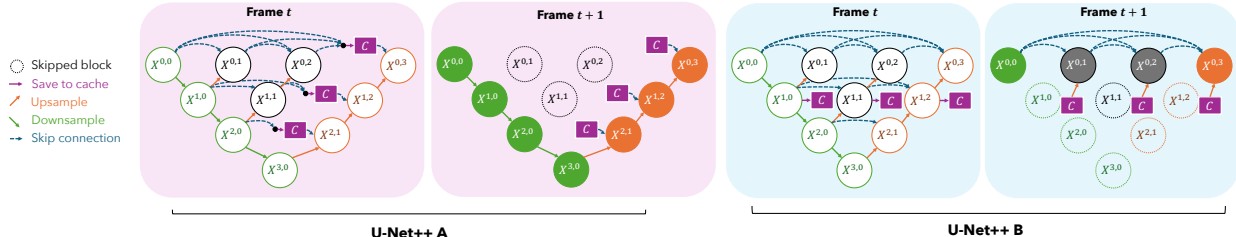

*Figure 11.* Ablation configurations for ExtraNet U-Net architecture.

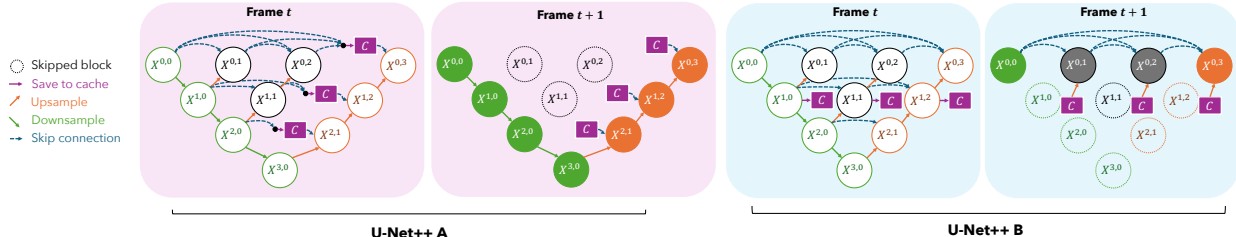

*Figure 12.* Ablation configurations for Implicit Depth U-Net++ architecture.

Table 7 shows the detailed settings of the cache refresh policies used in the ablation study.

| Configuration | Delta_H | Delta_L | Motion Vector | Non-Linear |
|---|---|---|---|---|
| **Parameters** | $\tau = 0.20$ | $\tau = 0.25$ | $\tau = 1$ | $c = 110, p = 1.4$ |

*Table 7.* Detailed settings of cache refresh policies. $\tau$ values are thresholds for SMAPE and average motion. $c$ and $p$ are parameters for the non-linear policy as described in DeepCache (Ma et al., 2024).

### A.4. Comparison to Video Games

We compare the amount of motion in our test set to those commonly observed in videos of real-world game play from GamingVideoSET (Barman et al., 2018) and CGVSD (Zadtootaghaj et al., 2020). These datasets are designed to evaluate

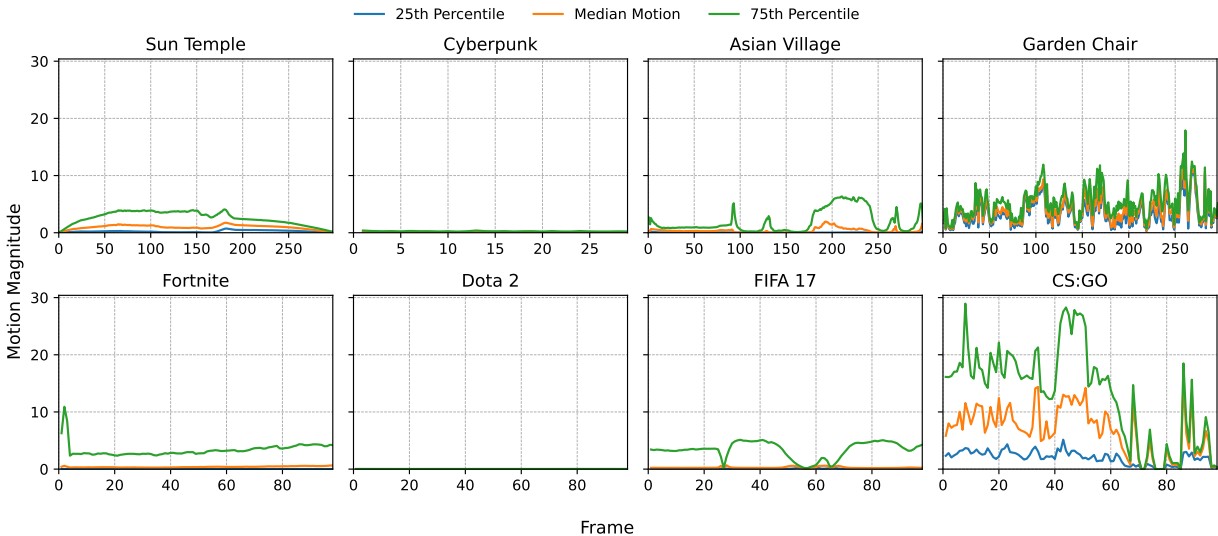

*Figure 13.* Optical flow motion comparison matching our test scenes to real-world game play from GamingVideoSET (Barman et al., 2018) and CGVSD (Zadtootaghaj et al., 2020).

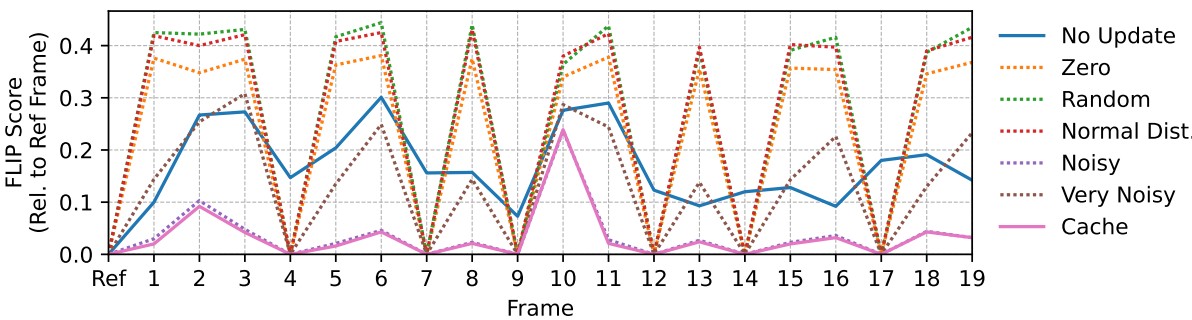

*Figure 14.* Null hypothesis test with Sun Temple on ExtraNet.

video compression algorithms and contain a representative variety of video game content. We measure the per-pixel deltas between each subsequent frame, as we use this metric to trigger cache refreshes. We also measure optical flow between frames as a secondary check that our test set exhibits a similar range of motion to real gameplay scenarios. Table 8 shows that our test set demonstrates a variety of motion patterns, within a similar range to the gaming video datasets.

### A.5. Null Hypothesis

We test the null hypothesis that the contents of the cache are irrelevant to the final result, meaning that the cached features can be replaced with any arbitrary value without affecting the network output. Specifically, we try replacing the cache with all zero values (Zero), uniformly random values (Random), normally distributed random values (Normal Dist.), all of which generate significantly worse results according to the FLIP score. Notably, the results are worse than not running the network at all (No Update).

We also test variations that add noise to the cached features. A small amount of noise (one standard deviation) does not affect the results, implying that the cached features are not sensitive to small perturbations. However, larger amounts of noise do degrade the final results. Figure 14 shows the results of the null hypothesis test for the Sun Temple scene on ExtraNet.

| Dataset | Scene | Per-Pixel Delta (Average) | Per-Pixel Delta (25th percentile) | Per-Pixel Delta (Median) | Per-Pixel Delta (75th percentile) | Average Optical Flow Magnitude |
|---|---|---|---|---|---|---|
| GamingVideoSET | CS:GO | 12.85 | 2.00 | 5.79 | 14.58 | 11.23 |
| GamingVideoSET | Diablo III | 2.73 | 0.71 | 1.41 | 3.08 | 2.35 |
| GamingVideoSET | Dota 2 | 0.99 | 0.00 | 0.00 | 0.71 | 0.19 |
| GamingVideoSET | FIFA 17 | 5.51 | 0.71 | 1.50 | 3.24 | 0.61 |
| GamingVideoSET | H1Z1 | 7.97 | 1.00 | 3.00 | 8.40 | 7.28 |
| GamingVideoSET | Hearthstone | 0.45 | 0.00 | 0.00 | 0.00 | 0.19 |
| CGVDS | Overwatch | 8.19 | 0.88 | 2.60 | 8.82 | 3.67 |
| CGVDS | Fortnite | 7.85 | 1.28 | 3.38 | 8.98 | 2.48 |
| Ours | Sun Temple | 10.20 | 0.00 | 2.55 | 10.20 | 1.81 |
| Ours | Cyberpunk | 10.73 | 0.99 | 2.37 | 7.45 | 0.36 |
| Ours | Asian Village | 17.85 | 2.55 | 7.65 | 22.95 | 5.80 |
| Ours | Garden chair | 40.89 | 9.71 | 23.07 | 51.47 | 4.53 |

*Table 8.* Video motion analysis comparison of per-pixel deltas and optical flow between our test sequences and real-world gaming video from GamingVideoSET (Barman et al., 2018) and CGVDS (Zadtootaghaj et al., 2020) datasets.

| Network | Cache size | Saved input size (for frame deltas) | Peak memory usage (baseline) | Peak memory usage (ReFrame) |
|---|---|---|---|---|
| ExtraNet | 21MB ($24\times360\times640$ `float32` tensor) | 21MB warped image ($6\times720\times1280$ `float32`) | 787 MB | 787 MB |
| FBSR | 158MB ($16\times540\times960$ `float32` temporal feature + $64\times540\times960$ `float32` HR feature) | 1MB LR feature ($3\times270\times480$ `float32`) + 24MB prev. result ($3\times1080\times1920$ `float32`) | 5.1GB | 5.3 GB |
| Implicit Depth | 84MB (seven $64\times192\times256$ `float32` tensors) | 2MB input image ($3\times384\times512$ `float32`) | 396 MB | 457 MB |

*Table 9.* Cache memory usage of our selected workloads. The cache size is determined by the size of the tensors stored between sequential frames. Peak memory usage is measured for the end-to-end network inference.

## A.6. Memory Consumption

ReFrame has a relatively small memory footprint, requiring storage of only a few tensors. The storage size is dependent on the network architecture and the resolution of the input image. Although high resolution images require a larger cache, the high resolution would also result in more FLOPs reduction with our technique, which helps justify the larger memory consumption. Table 9 measures the cache memory usage of our selected workloads. If memory is a large concern, the cache can also be stored in a lower precision, such as `float16`, with negligible changes in per-frame quality.

## A.7. Additional Results

We provide additional results for longer frame sequences in Table 10. Our main test set already contains several cache refreshes, which captures the latency and image quality effects of applying ReFrame. Longer sequences simply include a more varied set of cache refreshes, which does not significantly change the results.

We also provide additional results reporting the latency of the network inferences. Table 11 compares the latency with DeltaCNN, ReFrame, and both techniques combined against the baseline network. Applying ReFrame in addition to DeltaCNN compounds the latency reduction.

Table 12 reports the average latency and the worst case latency represented by the 95th percentile. As explained in Section 5, ReFrame is only effective at reducing the average latency.

| Workload | Scene | Skipped Frames ↑ | Eliminated Enc-Dec FLOPs ↑ | Speedup ↑ | FLIP ↓ | SSIM ↑ | PSNR ↑ | LPIPS ↓ | MSE ↓ |
|---|---|---|---|---|---|---|---|---|---|
| FE | Sun Temple (Long) | 45% | 24% | 1.25 | 0.025 | 0.990 | 36.01 | 0.011 | 4.31 |
|    | Asian Village (Long) | 69% | 37% | 1.50 | 0.037 | 0.987 | 39.55 | 0.018 | 4.85 |
| SS | Asian Village (Long) | 45% | 32% | 1.33 | 0.070 | 0.950 | 49.37 | 0.050 | 15.40 |

*Table 10.* Performance and image quality results with longer frame sequences. (FE - frame extrapolation, SS - supersampling)

| Frame | Baseline Latency | | Latency w/ DeltaCNN | | Latency w/ ReFrame | | Latency w/ DeltaCNN + ReFrame | |
|---|---|---|---|---|---|---|---|---|
| 0 - Ref | Full Inf. | 4.6 ms | Full Inf. | 4.6 ms | Full Inf. | 4.7 ms | Full Inf. | 4.7 ms |
| 1 - Delta | Full Inf. | 4.6 ms | Delta Inf. | 3.8 ms | Cached Inf. | 2.0 ms | Delta + Cached Inf. | 1.9 ms |
| 2 - Delta | Full Inf. | 4.7 ms | Delta Inf. | 3.7 ms | Full Inf. | 4.8 ms | Delta Inf. | 3.7 ms |

*Table 11.* Latency breakdown of three sequential frames of the Asian Village scene with ExtraNet comparing baseline network to DeltaCNN, ReFrame, and the combined approach of DeltaCNN + ReFrame.

| Workload | Scene | Baseline Avg. | Baseline 95pct. | Delta_L Avg. | Delta_L 95pct. | Delta_H Avg. | Delta_H 95pct. |
|---|---|---|---|---|---|---|---|
| FE | Sun Temple | 4.60 ms | 4.75 ms | 2.67 ms | 4.55 ms | 3.24 ms | 4.65 ms |
|    | Cyberpunk | 3.56 ms | 4.22 ms | 2.39 ms | 3.46 ms | 2.89 ms | 3.80 ms |
|    | Asian Village | 4.16 ms | 4.72 ms | 2.68 ms | 4.64 ms | 3.36 ms | 4.64 ms |
| SS | Sun Temple | 68.6 ms | 69.0 ms | 35.4 ms | 68.0 ms | 51.8 ms | 68.9 ms |
| IC | Garden Chair | 109 ms | 118 ms | 91 ms | 109 ms | 104 ms | 116 ms |

*Table 12.* Latency results of our selected workloads, measured in milliseconds. Avg. is averaged over all frames in the test set; 95pct. is the worst case latency at the 95th percentile.

