# OpenReview forum: "ReFrame: Layer Caching for Accelerated Inference in Real-Time Rendering"
_ICML.cc/2025/Conference — ICML 2025 poster_

### Official Review · Reviewer_axt7 · 2025-03-12

**Overall Recommendation:** 2

**Summary:**

This paper incorporates traditional caching methods previously used in U-Net based diffusion models into modern real-time rendering applications. The authors propose a novel caching policy that leverages motion vectors in graphics rendering pipeline to adaptively perform cache updates when the difference between frames has exceeded a preset threshold. The experiment results demonstrate that the proposed layer caching technique can accelerate several real-time rendering workloads with a speedup of up to 1.85x.

## update after rebuttal

Thank you for the rebuttal and the additional material that demonstrates the degree of motion across different evaluation datasets. However, I think the limitation to resource constrained devices and low-motion scenes considerably impacts the novelty of this work, as such environments are less sensitive to latency issues compared to dynamic scenes. In addition, one main contribution of the work is the adaptive policy, which distinguishes the work from prior uniform caching policies. However, I find the evaluation datasets not comprehensive and representative enough to demonstrate the effectiveness of the adaptive policy. There were only two videos provided during rebuttal, and the duration (number of frames) of the SunTemple scene is too short, which fails to demonstrate the superiority of an adaptive policy over a uniform policy. Furthermore, from the analysis of the optical flow measurements, the evaluation should incorporate more datasets exhibiting motion characteristics and temporal duration at least comparable to the AsianVillage scene. To this end, I would keep my original score.

**Claims And Evidence:**

Most claims made in the paper are supported by clear and convincing evidence. However, the authors claim in Section 2.3 that “Real-time rendering networks commonly take advantage of U-Net (Ronneberger et al., 2015) and U-Net++ (Zhou et al., 2018) architectures …”. To my knowledge, there are also recent works that propose transformer-based networks in real-time rendering applications, which the paper does not mention. It is essential to either mention the shortcomings of the transformer-based methods and provide relevant references or demonstrate that U-Net is indeed the mainstream SOTA method with more supportive materials.

**Essential References Not Discussed:**

References are sufficient.

**Experimental Designs Or Analyses:**

The experimental designs and analyses are sound to me.

**Methods And Evaluation Criteria:**

The evaluation criteria mostly follow the evaluations from prior work. However, there is limited discussion and/or demonstration of the dynamics of the scenes selected for evaluation, given that the effectiveness of the work largely depends on the degree of camera/object motion present in the scene.

**Other Comments Or Suggestions:**

* The metrics in the evaluation tables are not intuitive to read. For example, Table 1 states that the results are relative to the baseline, but does not clearly state how the presented FLIP, SSIM, PSNR, LPIPS, MSE values are calculated (does the baseline and experiment have a 41.41 PSNR and 0.0169 FLIP difference).
* (Minor) The topic of this work is better suited for computer graphics conferences, e.g., SIGGRAPH, EGSR.

**Other Strengths And Weaknesses:**

Strengths:
* The technique is training-free and does not require prior knowledge of the workload.
* The proposed adaptive caching policy is novel and leverages existing information in the rendering pipeline (motion vector) and requires no additional storage overhead.

Weaknesses:
* The caching technique’s effectiveness largely depends on the dynamic/motion of the scene.
* The work is limited to U-Net-like network architectures. There is also no mention of other types of networks (e.g. transformer-based models) in real-time rendering applications.

**Questions For Authors:**

* How does the technique perform in scenes with rapid camera movement and/or complex animations, which is common in many real-time latency-sensitive rendering applications such as gaming? In a highly dynamic scene where both the camera and the objects in the scene experience large degree of motion, it seems like the frequent cache updates may instead cause lag, as Figure 7 demonstrates that many frames with cache updates have longer inference time.
* In Section 3.1, it is mentioned that the caching scheme can be applied to networks beyond U-Net and U-Net++ architectures. Can you potentially adapt it to work with transformer-based models as well?
* In Section 4.2.1, it is claimed that the caching scheme is most effective on frame extrapolation tasks where the U-Net dominates inference time. Why is it that in Table 1, for Delta_L, Supersample instead achieved the highest speedup among the three workloads?

**Relation To Broader Scientific Literature:**

The proposed adaptive layer caching technique for real-time rendering is relevant to layer caching in U-Net based diffusion models.

**Theoretical Claims:**

There’s no proof for theoretical claims.

---

> ### Author Rebuttal · Authors · 2025-04-01
>
> Thank you for the detailed and thoughtful review. We appreciate the advice for Table 1 and will update accordingly. The baseline is used as the ground truth reference and the metrics are computed compared to this reference (i.e. MSE = $\frac{1}{N_{pixels}} \sum_{i=0}^{N_{pixels}-1} (pixel^{baseline}_i - pixel^{test}_i)^2$)
>
> To address reviewer concerns:
>
> *Cache Updates*
>
> With an adaptive policy, the cache would not be active during periods of rapid camera movement or complex animations that take up large portions of the screen space. However, this type of high activity is not typically sustained for a long period of time. In a video game example, there is likely periods of small motion while the players scanning the scene or hiding, interleaved with periods of large motion when a fight occurs. Our caching technique is most effective during small motions and we verify on gameplay recordings from GamingVideoSET [1] and CGVDS[2] that this behaviour exists for many real games.
>
> Furthermore, we believe the "lag" to refresh the cache is very small compared to the full inference time and mostly falls within the range of noise fluctuations of the full inference time. The lag is also primary caused by computing deltas rather than refreshing the cache. If this lag is a concern, switching to the N-5 policy nearly eliminates it. Storing the cache only adds around 0.02-0.05ms in latency.
>
> *Limitation to U-Net*
>
> Our proposed method indeed is designed for U-Net-like networks. We appreciate the advice to bring more attention to our current limitations and will update our paper accordingly.
>
> Although there are many transformer-based models used in real-time rendering applications (notably in NVIDIA DLSS 4.0), we believe U-Net-like convolutional networks are still heavily employed and more feasible to execute on lower-end devices. For example, super resolution in the Meta Quest VR headsets still relies on traditional algorithmic methods. Our caching technique is particularly useful in these lower-end devices where the proposed trade-off of slight quality loss for performance gains is valuable.
>
> We have not evaluated our technique on any transformer-based model. However, our method *can* be applied in a more general setting where there is a concatenation of extracted features, as demonstrated in the supersampling workload. If a network uses transformer-based feature extractors, then concatenate the features, our caching technique should still apply.
>
> *Highest Speedup*
>
> We apologize for the misleading text, which we will fix. In general, we expect the caching scheme to be most effective when the skipped computations contribute to a large part of the overall network inference. For ExtraNet, the U-Net dominates inference time, resulting in good speedup. Supersample, although not U-Net based, uses a concatenation that combines three major components of the network, two of which we cache. Therefore, Supersample actually skips more of the overall network, resulting in its high speedup.
>
> *Scene Motion*
>
> We measure "motion" to the best of our understanding as the average optical flow magnitude over our test scene frame sequences. As a comparison point, we also include this metric measured on some GamingVideoSET [1] and CGVDS[2] recordings.
>
> | **Dataset** | **Scene**    | **Average Optical Flow Magnitude** |
> | ----------- | ------------ | ---------------------------------- |
> | [1]         | CSGO         | 11.23 |
> | [1]         | Diablo III   | 2.35 |
> | [2]         | Overwatch    | 3.67 |
> | [2]         | Fortnite     | 2.48 |
> | Ours        | SunTemple    | 1.81  |
> | Ours        | CyberPunk    | 0.36  |
> | Ours        | AsianVillage | 5.80    |
> | Ours        | Garden chair | 4.53 |
>
> We have also uploaded plots of optical flow magnitude per frame over time as a profile for each of our test scenes here:
> https://drive.google.com/drive/folders/1hupA8P0ya11EFftxQAtL2RgMghG6XWrZ?usp=sharing
>
> *Supplementary Video*
>
> Please find sample videos of our workloads with and without the cache applied here: https://drive.google.com/drive/folders/1pQHb7kgL4Dy6rMyTIM7g1JnNpFB9koI4?usp=sharing
>
> *ExtraNet_Baseline_AsianVillage.mp4* shows the output from the baseline network (reference). *ExtraNet_Cache_AsianVillage.mp4* shows the output with our caching technique applied.
>
> *Supersampling_Baseline_HR_SunTemple.mp4* shows the output from the baseline network (reference).  *Supersampling_Cache_HR_SunTemple.mp4* shows the output with our caching technique applied.
>
>
> References:
>
> [1] N. Barman, S. Zadtootaghaj, S. Schmidt, M. G. Martini and S. Möller. 2018. "GamingVideoSET: A Dataset for Gaming Video Streaming Applications," 16th Annual Workshop on Network and Systems Support for Games (NetGames)
>
> [2] S. Zadtootaghaj, S. Schmidt, S. S. Sabet, S. Möller, and C. Griwodz. 2020. "Quality estimation models for gaming video streaming services using perceptual video quality dimensions," In Proc. of the 11th ACM Multimedia Systems Conference (MMSys '20)

---

### Official Review · Reviewer_p743 · 2025-03-12

**Overall Recommendation:** 3

**Summary:**

The paper proposed a training-free intermediate features caching method to accelerate diffusion model inferencing in real-time rendering. Targeting encoder-decoder style networks, the proposed method caches intermediate network layer outputs to be reused in subsequent inferences in order to reduce frame rendering latency.  Different caching policies are explored in the paper, including different cache refresh referencing methods.  The paper claims to achieve a speedup of 40% on average with negligible quality loss in three real-time rendering workloads.

**Claims And Evidence:**

The claims are well supported by extensive experiment results.

**Essential References Not Discussed:**

The related works listed the paper sufficiently covers the content for understanding the paper, including the tnter-frame similarity nature of real-time rendering and real-time rendering networks.

**Experimental Designs Or Analyses:**

The choices of workloads and cache refresh referencing methods are solid enough for designing extensive experiments.

**Methods And Evaluation Criteria:**

The methods and benchmark baseline make sense for evaluating the performance of the proposed method.

**Other Comments Or Suggestions:**

In Table 1, it would be clearer to list the type of workloads rather than the name of them.

**Other Strengths And Weaknesses:**

Strengths:
- The results show a great improvement in the rendering latency with negligible quality degradation.
- The paper thoroughly investigates the possibility of different caching policies.

Weaknesses:
- The test sets are slightly inadequate for evaluating different rendering scenes.
- The overhead of caching is yet to be discussed.

**Questions For Authors:**

- How is the overhead (i.e., memory) of the proposed method? Will scenes with higher resolution cause a huge occupation of system or GPU memory?
- Has a dynamic sensitivity been considered? The experiment results show very different performance levels; a fixed sensitivity may not be the best choice in practical usage in various scenes.

**Relation To Broader Scientific Literature:**

The key contributions of the paper extend caching methods, which are originally used diffusion models in to real-time rendering area.

**Theoretical Claims:**

The theoretical claims of layer caching for U-Net and U-Net++ in the paper are checked to be correct.

---

> ### Author Rebuttal · Authors · 2025-04-01
>
> We thank the reviewer for the helpful comments and will update Table 1 as suggested.
>
> Addressing reviewer questions:
>
> *Memory overhead:*
>
> Scenes with a higher resolution will require more memory. However, the cache only stores values of one or a few tensors, which will not cause a huge occupation of system or GPU memory. Our test scenes of 1080p is a typical input resolution. A higher resolution would also result in more FLOPs reduction by our technique, which helps justify the larger memory consumption.
>
> We add additional data measuring the memory footprint of the cache for each workload and peak memory usage during inference:
>
> | **Network**    | **Cache Size**                                               | **Peak Memory Usage (Baseline)** | **Peak Memory Usage (Caching)** |
> | -------------- | ------------------------------------------------------------ | -------------------------------- | ------------------------------- |
> | ExtraNet       | 21MB (24$\times$360$\times$640 `float32` tensor)             | 787 MB                           | 787 MB                          |
> | Supersample    | 158MB (16$\times$540$\times$960 `float32` temporal feature + 64$\times$540$\times$960 `float32` HR feature) | 5.1GB                            | 5.3 GB                          |
> | Implicit Depth | 84MB (seven 64$\times$192$\times$256 `float32` tensors)      | 396 MB                           | 457 MB                          |
>
> Peak memory usage is measured with `torch.cuda.max_memory_allocated()`. For ExtraNet, the peak memory usage occurs at a stage in the overall network that does not use our caching technique, thus the peak usage is unaffected by the caching scheme.
>
> If memory is an important concern, the cache can also be stored in a lower precision, such as `float16`, with negligible changes in per-frame quality.
>
> *Dynamic Sensitivity*
>
> We do not currently consider dynamic sensitivity. However, this is an interesting direction to potentially investigate. For our current work, we believe that dynamic sensitivity is not necessary. The sensitivity setting directly impacts the output quality, which should preferably stay consistent within one scene. The sensitivity settings can be tuned differently for each scene based on its contents.

---

### Official Review · Reviewer_7Mb7 · 2025-03-13

**Overall Recommendation:** 2

**Summary:**

The paper proposes to speed up real-time rendering tasks by leveraging the feature caching technique proposed in DeepCache, with extension to UNet++ and adaptive cache polices that are more suitable in the rendering context. Results show 1.4x speed up on average with negligible quality loss.

## update after rebuttal

I keep my sore for the reasons explained in the rebuttal comment

**Claims And Evidence:**

There are some claims that need further clarification, which I will discuss below under Experimental Designs Or Analyses.

**Essential References Not Discussed:**

To the best of my knowledge the essential related works are cited.

**Experimental Designs Or Analyses:**

Here are three key points that need further clarification:

1. Each scene contains only 10–20 frames for testing, which seems insufficient to capture the variety of motions and changes in adjacent frames. For example, the frames would be very similar to each other if there are only 10 frames, in which case the uniform cache policy may perform just as well as the proposed adaptive cache policy. ExtraNet, in comparison, uses 6000 frames for training and 1000 frames for testing. This is a significant difference, raising concerns about whether the proposed method has been evaluated on a sufficiently diverse test set. I expect the authors to provide an explanation for this choice and discuss its potential impact on the results.

2. For the image quality metrics in table 1, the authors only show the results for two variants of the proposed method, but not the results for baselines for comparison. How do the baselines perform in terms of those image quality metrics?

3. In section 4.2 the authors say "all scores are generally below the acceptable losses observed in other neural rendering systems, which report scores between 0.05 and 0.28 in their final results". Does the "score" here mean the FLIP score? Additionally, could the authors elaborate on why comparisons are made with neural rendering systems (specifically, the three cited papers), which seem to be addressing different tasks?

4. In the supersampling task, what's the original scaling factor used in the baseline's experiments? Is it also 4 times?

**Methods And Evaluation Criteria:**

They make sense. The method is very straightforward, and the evaluation criteria are well-known in the literature.

**Other Comments Or Suggestions:**

-

**Other Strengths And Weaknesses:**

-

**Questions For Authors:**

The proposed method is simple yet effective. However, I remain unconvinced due to the concerns raised under Experimental Designs or Analyses. I would appreciate it if the authors could provide a more detailed explanation and justification addressing these issues.

**Relation To Broader Scientific Literature:**

This paper has the potential to accelerate various components of the modern rendering pipeline, such as frame extrapolation, supersampling, and image composition, as demonstrated in the paper. The proposed method could be further extended to video processing tasks, such as video generation.

**Theoretical Claims:**

There is no theoretical claim in the paper.

---

> ### Author Rebuttal · Authors · 2025-04-01
>
> Thank you for the helpful review and feedback on our paper.
>
> To clarify key points:
>
> 1. *Testing sequence length:*
> We tested short sequences because the cache is only valid for a few frames, and the test of 10-20 frames already includes several cache refreshes. However, we add additional data for both the Asian Village scene and Sun Temple scene that tests sequences of 250 frames. We find that the speedups and quality results for 250 frames are aligned with our original experiments, with a more varied pattern of cache refreshes under an adaptive policy.
>
> | **Workload** | **Scene**     | **Full Frames** | **Enc-Dec FLOPs** | **Speedup** | **FLIP** | **SSIM** | **PSNR** | **LPIPS** | **MSE** |
> | ------------ | ------------- | --------------- | ----------------- | ----------- | -------- | -------- | -------- | --------- | ------- |
> | ExtraNet     | Asian Village | 31%             | 63%               | 1.50        | 0.037    | 0.987    | 39.55    | 0.018     | 4.85    |
> |              | Sun Temple    | 55%             | 76%               | 1.25        | 0.025    | 0.990    | 36.01    | 0.011     | 4.31    |
> | Supersample  | Asian Village | 55%             | 68%               | 1.33        | 0.070    | 0.950    | 49.37    | 0.050     | 15.40   |
>
> Furthermore, we investigated the quality of our test sets by comparing to examples from GamingVideoSET [1] and CGVDS[2], which are datasets of real gameplay video recordings. We compare the distribution of our frame-to-frame pixel deltas to those observed in the gaming datasets to judge if our videos are a reasonable representation of motion that occurs in video games. We find that our distribution aligns well with GamingVideoSET [1] and CGVDS[2], matching better to first-person perspective games.
> | **Dataset** | **Scene**    | **Per-Pixel Delta (Average)** | **Per-Pixel Delta (25th percentile)** | **Per-Pixel Delta (Median)** | **Per-Pixel Delta (75th percentile)** |
> | ----------- | ------------ | ----------------------------- | ------------------------------------- | ---------------------------- | ------------------------------------- |
> | [1]         | CSGO         | 12.85                         | 2.00                                  | 5.79                         | 14.58                                 |
> | [1]         | Diablo III   | 2.73                          | 0.71                                  | 1.41                         | 3.08                                  |
> | [2]         | Overwatch    | 8.19                          | 0.88                                  | 2.60                         | 8.82                                  |
> | [2]         | Fortnite     | 7.85                          | 1.28                                  | 3.38                         | 8.98                                  |
> | Ours        | SunTemple    | 10.20                         | 0.00                                  | 2.55                         | 10.20                                 |
> | Ours        | CyberPunk    | 10.73                         | 0.99                                  | 2.37                         | 7.45                                  |
> | Ours        | AsianVillage | 17.85                         | 2.55                                  | 7.65                         | 22.95                                 |
> | Ours        | Garden chair | 40.89                         | 9.71                                  | 23.07                        | 51.47                                 |
>
> 2. *Image quality metrics:*
>
> The reported image quality metrics are reference-based and show the relative quality of our proposed method when compared to the baseline image without our method. For example, the baseline image would have MSE = 0 against itself. We are not concerned with the orthogonal measure in quality of the baseline network since we are not targeting quality improvements, only latency and computational reduction.
>
> 3. *FLIP score:*
>
> Unfortunately, there is no generally agreed upon threshold of “acceptable” quality loss for FLIP. The FLIP score purely measures perceptual quality differences between two rendered images and is not influenced by the rendering method. Therefore, we included a range of scores from other neural rendering papers that implicitly claim their scores as “acceptable” in order to help readers interpret and judge our scores.
>
> 4. *Supersampling:*
>
> Yes, the baseline scaling factor is always 4x in our results.
>
>
>
> [1] N. Barman, S. Zadtootaghaj, S. Schmidt, M. G. Martini and S. Möller, "GamingVideoSET: A Dataset for Gaming Video Streaming Applications," *2018 16th Annual Workshop on Network and Systems Support for Games (NetGames)*
>
> [2] S. Zadtootaghaj, S. Schmidt, S. S. Sabet, S. Möller, and C. Griwodz. 2020. "Quality estimation models for gaming video streaming services using perceptual video quality dimensions," In Proc. of the 11th ACM Multimedia Systems Conference (MMSys '20)

---

> > ### Comment · Reviewer_7Mb7 · 2025-04-09
> >
> > Thank you for the rebuttal. However, I remain unconvinced by the response to the first key point, which is also my primary concern. Specifically, my concern was not about the comparison with and without caching, as addressed in the rebuttal, but rather about the comparison between uniform and adaptive caching. Since adaptive caching is presented as one of the main contributions of the paper—and given that the concept of layer caching itself is not novel—this distinction is important. As I mentioned in my review, a uniform caching policy might perform just as well as the proposed adaptive policy when the number of frames is limited, hence it's crucial to compare them in the setting with more frames.
> >
> > Regarding the supersample workload, the authors present results on the Sun Temple scene in the main paper, so it would be more convincing to provide results on the same scene with more frames in the rebuttal, besides the Asian Village scene. Additionally, since the baseline uses 1000 frames for testing, it would be more appropriate to show results using 1000 frames—at least for the ExtraNet workload—to allow for a fair comparison.
> >
> > Finally, there is an apparent performance drop as the number of frames increases from 10 to 250, which raises concerns about the method’s scalability and effectiveness on longer video sequences in real-world scenarios. Given these, I would keep my score.

---

### Official Review · Reviewer_PRfu · 2025-03-13

**Overall Recommendation:** 3

**Summary:**

The authors propose a technique for caching intermediate features of U-net style networks to skip computation of hidden layers. These intermediate features are recomputed when the changes is above a certain threshold. Overall, this produces an average improvement of performance without significant drop in quality on three tasks (frame extrapolation, neural supersampling and image composition).

## Update after rebuttal
The author's rebuttal arguments have addressed my concerns adequately. I've updated my rating for this review.

**Claims And Evidence:**

The proposed method caches intermediate feature maps and leveraging temporal coherence in U-net style networks to gain performance in rendering applications without significant drop in quality. Various caching policies are compared.

**Essential References Not Discussed:**

None that I'm aware of.

**Experimental Designs Or Analyses:**

The overall experimental design makes sense and accounts for the total number of full inferences, floating point operation decreases, average speedup and quality measurement. An additional metric that might be relevant for this article might be the 95th percentile of inference time.

**Methods And Evaluation Criteria:**

The method evaluates the average latency of the networks and also the quality of the rendered image against the full network evaluation using various image quality metrics like FLIP, LPIPS, SSIM and PSNR on a variety of tasks such as frame extrapolation, neural supersampling and image composition.

**Other Comments Or Suggestions:**

None

**Other Strengths And Weaknesses:**

The proposed technique is original and clearly explained. The technique while achieving an average speedup has slowdowns at irregular intervals to recompute the caches.

**Questions For Authors:**

1. The method proposed provides an average improvement in throughput of the framerate but it is unclear how a consistent framerate could be achieved which is important for rendering applications. Do you have any thoughts or ideas on how it could be achieved?
2. Do you leverage any CUDA specific memory or caching techniques using custom kernels or APIs?
3. What runtime do you observe when using DeltaCNN on the Asian village scene both with and without the proposed caching scheme?

**Relation To Broader Scientific Literature:**

The key contributions of the paper are directed towards the high throughput inference line of work that leverages a variety of techniques such as sparsity, network architectures, precision, etc to gain performance while maintaining quality.

**Theoretical Claims:**

No theoretical claims in the paper.

---

> ### Author Rebuttal · Authors · 2025-04-01
>
> Thank you for the helpful review and feedback on our paper.
>
> We have added additional data on the 95th percentile as suggested, reported as runtime in milliseconds.
>
> | **Workload**                  | **95th Percentile Latency (with cache)** | **95th Percentile Latency (baseline, no cache)** |
> | ----------------------------- | ---------------------------------------- | ------------------------------------------------ |
> | ExtraNet - SunTemple          | 4.55 ms                                  | 4.75 ms                                          |
> | ExtraNet - CyberPunk          | 3.46 ms                                  | 4.22 ms                                          |
> | ExtraNet - Asian Village      | 4.64 ms                                  | 4.72 ms                                          |
> | Supersample - SunTemple       | 68 ms                                    | 69 ms                                            |
> | Implicit Depth - Garden Chair | 109.2 ms                                 | 118.2 ms                                         |
>
> However, as the reviewer points out, our method improves the latency ***on average*** and therefore has little effect on the slowest percentiles. Relating to question (1), we concede that our method cannot maintain a consistently faster frame rate. Although rendering requires a consistent frame rate, our cache technique focuses on post-processing networks that enhance the rendering, rather than rendering the base image itself. We believe our method is most suitable in low-end devices where additional image quality improvements from neural networks are included in a best-effort manner. Furthermore, reducing computation on average still reduces energy consumption, which is an especially important target in mobile-class devices such as VR headsets.
>
> One possible solution to produce a more stable frame rate is to amortize the cache refresh over two (or more) frames. The first frame can continue utilizing the cache contents while starting a full inference to refresh the cache. The refresh completes in the second frame asynchronously without delaying the first frame and the second frame can use the updated cache.
>
> To answer question (2), we do not leverage any CUDA specific memory or custom kernels in our cache implementation. The cache is relatively small, storing one or several tensors ranging from 20-160MB in total for our experiments. Also, loading these tensors from the cache is not different from loading the intermediate results from the previous network layer in a regular full inference, thus we find it unnecessary to optimize this operation.
>
> We add DeltaCNN related runtime results to answer question (3), matching Table 3 in the paper. Our caching approach produces higher quality results in less time than DeltaCNN due to the nature of the workload as described in the paper (high channel dimension and lower sparsity). When combined, the runtime is not significantly improved, at detriment to the quality.
>
> | Baseline | Runtime (ms) | DeltaCNN        | Runtime (ms) | Cache            | Runtime (ms) | DeltaCNN + Cache         | Runtime (ms) |
> | -------- | ------------ | --------------- | ------------ | ---------------- | ------------ | ------------------------ | ------------ |
> | Ref      | 4.6          | Full Inference  | 4.6          | Full Inference   | 4.7          | Full Inference           | 4.7          |
> | Delta-1  | 4.6          | Delta Inference | 3.8          | Cached Inference | 2            | Delta + Cached Inference | 1.9          |
> | Delta-2  | 4.7          | Delta Inference | 3.7          | Full Inference   | 4.8          | Delta Inference          | 3.7          |

---

### Decision · Program_Chairs · 2025-05-01

**Decision:**

Accept (poster)

**Comment:**

The paper received four scores: two weak accepts and two weak rejects, placing it exactly on the borderline. On one hand, the reviewers praised the simple and efficient approach, which does not require training. The method achieves a 40% speed improvement with a negligible difference in quality. On the other hand, there were concerns about the evaluation setup (each scene contained only 10–20 frames for testing) and the use of a U-Net instead of Transformers. The authors provided some additional details and results addressing the former. Regarding the latter (U-Net vs. Transformers), while it is indeed important to consider recent architectures, the AC believes the authors’ response is adequate. We should adapt methods to new architectures, but we also need to start somewhere.

Given that even the negative reviews do not strongly recommend rejection and acknowledge the potential significance of the work, the AC has decided to accept the manuscript.